# Comparison of lower limb muscle activation between slow and fast tempos during the stepping test in young men

**Keiji Koyama**[ID]*, **Chisa Watanabe, Yusuke Oyama**

Toin University of Yokohama, Kanagawa, Japan

* koyakei@toin.ac.jp

## Abstract

The aim of this study was to determine whether and which lower limb muscles contribute to maintaining dynamic balance during the stepping test. Twelve healthy young men performed in-place stepping at tempos of 44 (slow) and 132 (fast) bpm. Vertical ground reaction forces and lower limb muscle activity were recorded using a force plate and a telemetric electromyography (EMG) system, respectively. EMG signals were recorded from the rectus femoris (RF), vastus lateralis (VL), biceps femoris (BF), gastrocnemius medialis (GM), and soleus (SOL) on the right leg. The single-leg support time and the time difference between metronome sound and foot contact at the slow tempo were greater than those at the fast tempo. The impact force at the fast tempo was greater than that at the slow tempo. In the single-leg support phase, the mean EMG signals of the RF, VL, and SOL at the slow tempo were 63%, 17%, and 23% lower, respectively, than those at the fast tempo, whereas the integrated EMG signals of the VL, BF, GM, and SOL at the slow tempo were 171%, 315%, 214%, and 157% greater, respectively. There were no significant relationships among the rate of change in the single-leg support time, the time difference from the slow to the fast tempo, or the rate of change in the EMG activity of each muscle. In the stepping test, movement characteristics associated with tempo were reflected in lower limb muscle activity, while dynamic balance indicators appeared to be influenced by factors other than muscle activation.

## Introduction

Dynamic stability has been evaluated as a predictor of mobility and fall risk in daily life for older adults [1,2], as an assessment measure for physical performance [3,4] and as a risk factor for injury prevention [5,6] for younger individuals. One method commonly used to assess dynamic stability is the stepping test. The stepping test was developed as a very simple and safe test that reflects the dynamic balance ability of a wider range of subjects, including young to middle-aged and older subjects,

**Data availability statement:** All relevant data are within the manuscript.

**Funding:** K.K. (Keiji Koyama) was supported by a Grant-in-Aid for Scientific Research from the Japan Society for the Promotion of Science (Grant-in-Aid for Young Scientists (19K20021) and Grant-in-Aid for Scientific Research (23K10674)). The funders had no role in study design, data collection and analysis, decision to publish, or preparation of the manuscript.

**Competing interests:** The authors declare that they have no known competing financial or personal interests that could have appeared to influence the work reported in this paper.

than other dynamic balance ability tests [7], such as the Berg Balance Scale [8], the Tinetti Performance-Oriented Mobility Assessment [9], the Timed "Up & Go" Test [10] and the single-leg landing test [11]. The stepping test is performed in place (i.e., without moving forwards) to a fixed tempo, and it is necessary to shift the centre of gravity (CoG) of the body to the left or right leg alternately, similar to walking. Single-leg and double-leg support times are measured during the stepping test, and the longer the single-leg support time and the shorter the double-leg support time are, the greater the dynamic balance ability [12]. Greater dynamic balance ability is also indicated by shorter differences between the tempo specified by the metronome and the timing of foot contact [12]. These variables of balance ability decrease with age [13]. On the basis of the characteristics of the stepping test, force generation by the lower limb muscles is necessary to smoothly shift the CoG of the body from side to side and to support a stable posture with only one-leg.

Humans maintain stable posture by generating the necessary forces through skeletal muscles on the basis of information from the central nervous system, including the vestibular, visual, somatosensory, and musculoskeletal systems [14]. In dynamic balance, muscle groups in the upper limbs, trunk, and lower limbs work cooperatively to maintain equilibrium by keeping the CoG of the body within the base of support according to movement speed [15,16]. In the stepping test, the base of support is adjusted in the direction of the step, enabling rapid and reflexive automatic compensatory step responses. For example, when the CoG of the body shifts outwards, a compensatory step response is typically executed inwards, thereby stabilizing posture. During this process, increased surface electromyography (EMG) activity is observed in both the distal and proximal lower limb muscles [17]. These muscle activity levels are modulated according to the magnitude of the displacement of the CoG of the body [18]. Winter [19] hypothesized that postural sway in the anterior–posterior direction is associated with ankle joint stability, whereas sway in the mediolateral direction is related to hip joint stability. Postural sway in the anterior–posterior direction is indeed linked to muscle activity in the ankle joint [20], whereas postural sway in the mediolateral direction is significantly influenced when the stabilizing muscles of the ankle joint are fatigued [21]. Therefore, comparing muscle activity in the ankle and hip joints during stepping tests with varying movement speeds may provide insights into the muscles required to maintain dynamic stability.

The amount of force generated by the lower limb muscles is important for maintaining posture stability. For example, a previous study has shown that the amount of foot muscle activity was amplified when the support base progressively narrowed from sitting to double-leg standing and single-leg standing to increase instability [22]. In another study, the base of the support surface in single-leg standing was smaller than that in double-leg standing, and muscle activity in the lower limbs increased to maintain balance due to increased instability [23]. Moreover, wearing shoes with a curved anterior and posterior sole, called unstable shoes, was found to increase the activation of extrinsic foot muscles during standing [24] and lower limb muscles during walking [25,26]. According to the findings of these previous studies, the need for force generation by lower limb muscles to maintain stability during physical exercise can be determined by whether

the activation of lower limb muscles is amplified when a subject is forced into an unstable posture. In the stepping test, both younger and older adults showed greater differences in the timing of the metronome tempo and the timing of foot contact at slower tempos than at faster tempos, resulting in difficulty in controlling postural stability [12]. However, the amount of lower limb muscle activity during the stepping test has not been measured, and it is unclear whether force generation by lower limb muscles is related to dynamic balance capacity in the stepping test. The mean EMG (mEMG) and integrated EMG (iEMG) signals respectively reflect the intensity of muscle activation and the cumulative muscular activity over time [27]. Given these properties, stepping tempo is expected to differentially influence mEMG and iEMG. Specifically, during slower tempo stepping, the movement speed is reduced and impact forces at landing are lower, which may result in decreased mEMG values. In contrast, the prolonged single-leg support time in the slow condition likely leads to increased iEMG values due to the sustained activation of the lower limb muscles. Based on these considerations, we hypothesized that stepping at slower tempos would increase integrated muscle activation due to prolonged single-leg stance. Furthermore, because dynamic balance is more challenging at slower tempos, the rate of change in dynamic balance indicators between tempo conditions is expected to correlate with the rate of increase in iEMG across the lower limb muscles. Elucidating the role of muscle groups in dynamic stability is essential for the development of effective training programmes aimed at increasing dynamic stability in young individuals as well as rehabilitation protocols for subjects following injuries. Therefore, the aim of this study was to measure the degree of change in the activation of lower limb muscles at different tempos in the stepping test and to determine whether and which lower limb muscles are important for maintaining dynamic balance in the stepping test.

## Materials and methods

### Subjects

To control for factors that influence dynamic stability, such as age [28], sex [29], and body composition [30], the subjects in this study were limited to 20- and 21-year-old men. Additionally, subjects with a body mass index (BMI) of less than 25 were selected on the basis of the World Health Organization (WHO) definition (normal: BMI < 25; overweight: 25 ≤ BMI < 30; and obese: BMI ≥ 30). Standing height and body mass were measured while the subjects were wearing lightweight clothes and were without shoes, and BMI was calculated as [BMI = body mass (kg)/height (m)・ height (m)]. Twelve healthy men (age, 20.4 ± 0.5 years; height, 1.75 ± 0.04 m; body mass, 63.9 ± 7.4 kg) volunteered to participate in this study. None of the subjects were taking medications or had peripheral nerve dysfunction, another neurological disorder, or any injury to the feet or legs within the past year. The physical activity levels of the subjects were assessed using the short form of the International Physical Activity Questionnaire (IPAQ), based on the frequency and duration of physical activities performed at three intensity levels (vigorous, moderate, and low) over the past seven days [31]. Total weekly physical activity was estimated by weighting the time spent at each intensity level with its corresponding metabolic equivalent (MET) energy expenditure. The MET values were set at 8.0, 4.0, and 3.3 for vigorous-, moderate-, and low-intensity activities, respectively [32]. The subjects' total MET score was 5633 ± 2522 METs, with all individuals meeting or exceeding the moderate-active threshold (600 ≤ METs < 3000) defined in a previous study [32]. This study was approved by the Toin University of Yokohama Human Research Ethics Committee (ethical approval number: I-29) and was performed in accordance with the guidelines of the Declaration of Helsinki. Subjects were prospectively recruited during the period from April 1 to May 31, 2023. All the subjects provided written informed consent before the onset of the study. Potential volunteers aged between 20 and 21 years were verbally recruited from a class at the local university during the fall semester. The experimental sample size was estimated on the basis of data from a previous study [12], in which the timing discrepancy between foot contact and metronome sound was measured using the stepping test as an indicator of dynamic balance ability. A significance level of less than 0.05 (Zα/2 (0.025) = 1.96) and a test power of 95% were used in the calculations. According to this calculation, the minimum required sample size was 11. To account for a potential 10% dropout rate due to measurement errors or subject withdrawal from the study, one additional subject was included. Therefore, the minimum required sample size was 12.

## Experimental procedure

Prior to data collection, the subjects completed a brief warm-up consisting of stepping, walking, and stretching exercises targeting the quadriceps, hamstrings, and triceps surae muscles. The stepping test was performed barefoot and in short training pants to minimize external influences on movement. The test was conducted in place (i.e., without forward loco-motion) at a fixed tempo. Dynamic balance ability during the stepping test varies with movement speed, requiring different motor control strategies. Specifically, at a slow tempo of 44 bpm, the single-leg support time increases, whereas at a fast tempo of 132 bpm, maintaining synchronization with the metronome becomes more challenging [28]. In young adults, the timing discrepancy between foot contact and the metronome sound, which serves as an indicator of instability, is greater at slower tempos than at faster tempos, resulting in increased instability [33]. In this study, the aim was to increase insta-bility by manipulating the movement speed during the stepping test in young adults and to investigate the effects of this manipulation under two tempo conditions, in which the roles of the lower limb muscle groups differ markedly.

The subjects stood with one foot on each of two force plates and performed stepping at either 44 bpm (slow tempo) or 132 bpm (fast tempo), as dictated by a metronome [28]. To minimize learning effects, a practice trial was conducted for each tempo prior to testing to familiarize subjects with the movement. During the practice trial, it was confirmed that subjects performed the stepping test with forefoot landing. After familiarization, the stepping test was conducted once for 20 seconds at each tempo. A one-minute rest period was provided between stepping tests at different tempos, and the order of tempo conditions was randomly assigned. Two force plates were used simultaneously to detect phases of double-leg and single-leg support. Vertical ground reaction force data were used to determine foot contact and toe-off events, enabling calculation of single-leg support time and the time difference between right foot contact and the metro-nome sound. During the single-leg support phase on the right leg, vertical ground reaction force and muscle activity were recorded using a force plate (Model 9286A, Kistler, Winterthur) and a telemetric EMG system (WEB-5000, Nihon Kohden, Tokyo), respectively. Analogue EMG signals, vertical ground reaction forces and metronome sound were simultaneously sampled at 1 kHz using a 12-bit analogue-to-digital converter and synchronized. Data were compared between the slow and fast tempo conditions. To eliminate potential bias due to limb dominance, only the dominant right leg was analysed for all subjects. Leg dominance was determined based on criteria used in previous studies [34,35], where the dominant leg was defined as the one meeting at least two of the following three conditions: (1) the leg used to kick a ball, (2) the leg used to step up stairs, and (3) the leg that steps forward to prevent a fall.

## Time parameters

The vertical ground reaction force and metronome sounds during the stepping test at 44 bpm are shown in Fig 1a. The single-leg support time of the right leg and the time difference between foot contact and metronome sound were calcu-lated [12]. The single-leg support time of the right leg is the duration for which the subject stands on only the right leg while stepping. The time difference was selected and defined as the difference between the subject's right foot contact time and the metronome sound at each tempo (Fig 1b). Subjects with a smaller time difference between foot contact and the metronome sound were found to synchronize with the tempo more accurately and to demonstrate superior dynamic balance ability [12]. These time parameters were calculated from twelve consecutive steps, and the single-leg support times and time differences for the right leg were represented as the mean values from three steps of right foot contact.

## Vertical ground reaction force

In this study, only the vertical component of the ground reaction force was measured to evaluate foot contact and toe-off timing, as well as vertical force-related variables during landing. Mediolateral and anteroposterior components were not recorded. The impact force at initial contact and the force required for step transitions of the right foot were assessed using vertical ground reaction force data under slow-tempo and fast-tempo conditions. Representative examples of

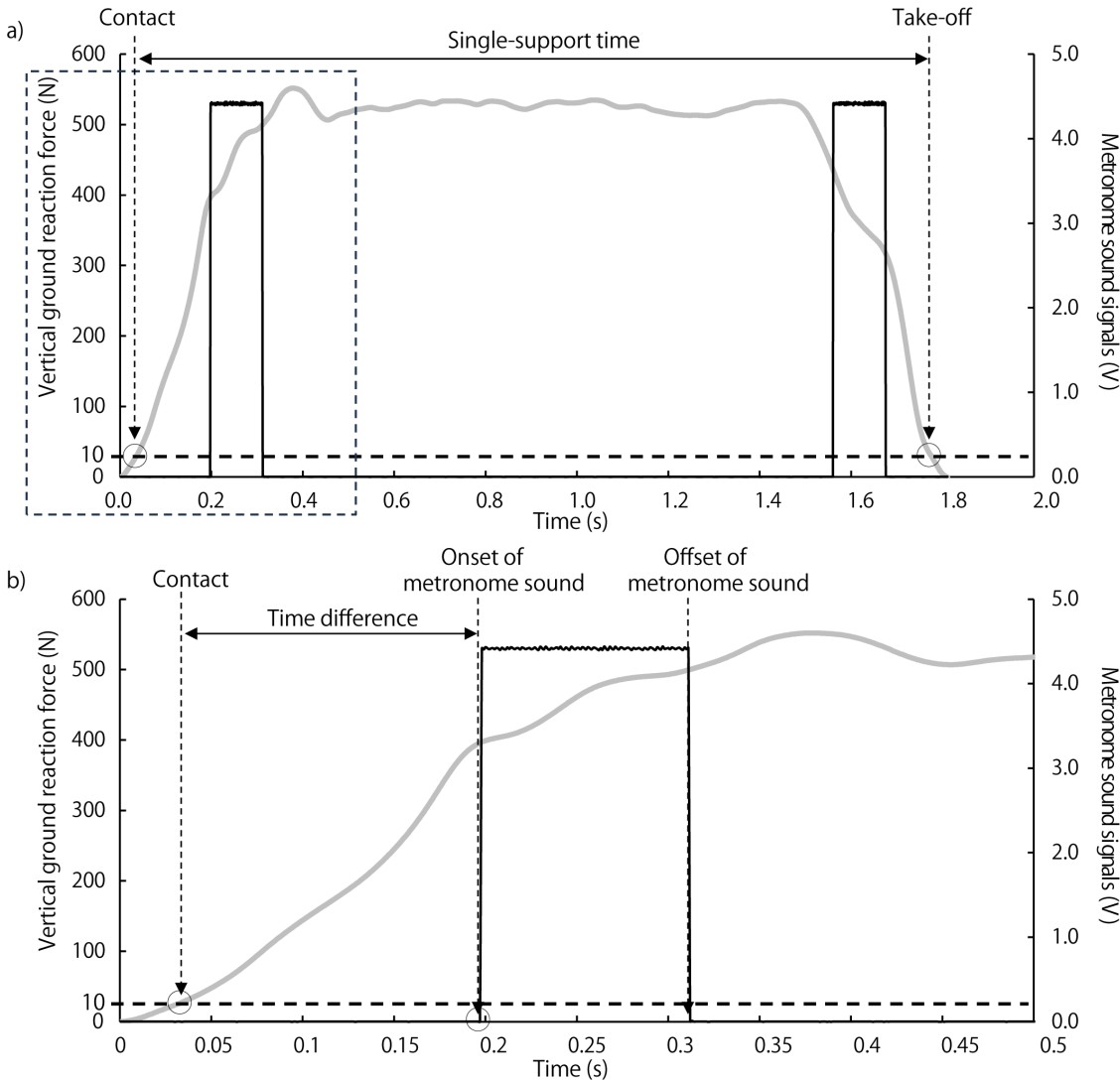

**Fig 1. Single-leg support time and time difference between metronome sound and foot contact. (a)** Typical vertical ground reaction force and metronome sound signals during the single-leg support phase, from right foot contact to take-off, at the slow tempo in the stepping test. **(b)** Time difference between foot contact and the metronome sound. The black rectangular waveform represents the metronome voltage signal, recorded at 1.36-s intervals (44 bpm) in the slow tempo condition. The grey continuous waveform represents the vertical component of the ground reaction force. Foot contact was defined as the point at which the ground reaction force exceeded 10 N, and take-off as the point at which it fell below 10 N. The duration from foot contact to take-off was defined as the single-leg support time, and the duration from foot contact to the onset of the metronome sound was defined as the time difference.

vertical ground reaction forces recorded under slow-tempo and fast-tempo conditions are shown in Fig 2a and 2b, respectively. Based on previous studies [36], the recorded raw vertical ground reaction force signals were processed using LabChart software (LabChart 8 for Windows, AD Instruments). Vertical ground reaction force data were low-pass filtered at 20 Hz using a zero-phase-lag finite impulse response (FIR) filter with a Kaiser window. Foot contact and toe-off during each step were defined using a 10 N vertical ground reaction force threshold [37]. The contact phase was divided into an early phase and a late phase on the basis of the midpoint of the contact time. The peak ground reaction force during the early phase was defined as the impact force, whereas the peak ground reaction force during the late phase was assessed

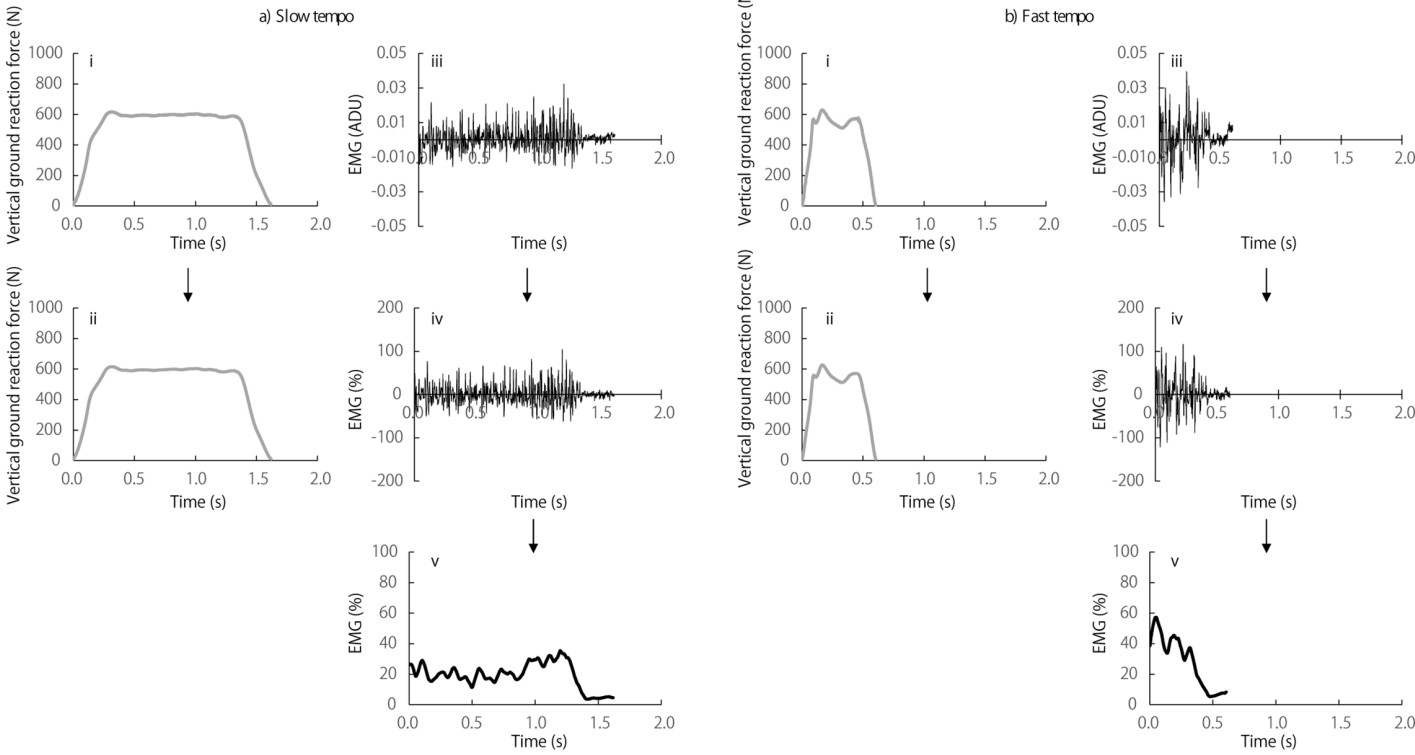

**Fig 2. Electromyography (EMG) and vertical ground reaction force signals during single-leg support. (a)** Typical vastus lateralis EMG and vertical ground reaction force signals during the single-leg support phase, from right foot contact to take-off, at the slow tempo in the stepping test. **(b)** Same signals at the fast tempo. The progression from raw to processed data is shown for ground reaction force (i, ii) and EMG (iii–v), respectively. i: raw ground reaction force data, ii: low-pass filtered at 20 Hz, iii: raw EMG data, iv: bandpass filtered (10–450 Hz) and normalized to maximal voluntary contraction, v: root mean square calculated using a 100-ms moving window.

as the force required for step transitions. Both the impact force and the force required for step transitions were normalized to each subject's body weight. The mean values of the impact force and the force required for step transitions were calculated across six steps and compared between the 44 bpm and 132 bpm conditions. These six steps were also used for timing and EMG analysis.

## EMG analysis

EMG signals were recorded from five superficial right leg muscles (rectus femoris, RF; vastus lateralis, VL; biceps femoris, BF; gastrocnemius medialis, GM; and soleus, SOL) using a telemetric EMG system at a sampling frequency of 1 kHz. Postural control in humans is regulated by the ankle joint in the anterior–posterior direction and by the hip joint in the medial–lateral direction [19]. The force generation of the muscle groups surrounding these joints is considered to contribute to dynamic stability. Indeed, during assessments of dynamic stability, EMG measurements have been specifically conducted on these muscles [38,39]. The signals were passed through a locally grounded preamplifier (AM-511H, Nihon Kohden, Tokyo, Japan) to a receiving unit (ZR-550H, Nihon Kohden, Tokyo, Japan). The analogue signals were telemetrically sent to a recording computer (a 12-bit AD converter). EMG signals were detected by bipolar electrodes (Blue Sensor M-00-S/50, Ambu, Ballerup, Denmark) with a 10 mm diameter and a centre-to-centre distance of 20 mm. Electrode placement followed the recommendations of the SENIAM project [40], including alignment parallel to the muscle fibres.

To ensure accurate electrode positioning, a B-mode ultrasound system (FC1-X, Fujifilm, Tokyo, Japan) with an electronic linear array probe (HFL38xp/13–6, 6–13 MHz) was used. The ultrasound probe was coated with a water-soluble transmission gel and placed perpendicular to the skin without compressing the dermal surface. After shaving and cleaning the skin with isopropyl alcohol, electrodes were placed midway along the muscle belly, parallel to the underlying muscle fibres, with care taken to avoid the borders of adjacent muscles to minimize crosstalk. The reference electrode was placed over the flat portion of the anteromedial aspect of the tibia. Pulling artifacts were avoided by securing the electrode cables to the skin using adhesive tape. After the skin was shaved and cleaned with isopropyl alcohol, the electrodes were placed midway along each muscle, parallel to the direction of the underlying muscle fibres. The reference electrode was attached over the flat portion of the anteromedial aspect of the tibia. Pulling artefacts were avoided by carefully fixing the electrode cables to the skin with tape. Based on a previous study [41], the recorded raw EMG signals were processed using Lab-Chart software (LabChart 8 for Windows, AD Instruments). The signals were first smoothed via bandpass filtering with a zero-phase-lag Finite Impulse Response type filter with a Kaiser window (cutoff frequency: 10–450 Hz), followed by root mean square (RMS) calculation using a 100 ms moving window. The mean EMG (mEMG) and integrated EMG (iEMG) signals were calculated during the single-leg support phase. The absolute mEMG and iEMG values were further normalized in each muscle with respect to the maximum value during maximum voluntary contraction (MVC); these normalized values are denoted as %mEMG and %iEMG, respectively. The subjects performed a 3-second isometric maximal voluntary contraction (MVC) for each muscle against manual resistance. Prior to the MVC trials, they practiced at a submaximal level. A 1-minute rest period was provided between trials to minimize the effects of fatigue. For the evaluation of MVCs in hip flexion for the rectus femoris (RF), knee extension for the vastus lateralis (VL), and knee flexion for the biceps femoris (BF), all subjects were seated on a quadriceps bench with their hips and knees flexed at 90 degrees. From this position, the RF was assessed through hip flexion MVC, while the VL and BF were evaluated through knee extension and knee flexion MVCs, respectively. For the evaluation of MVCs of the gastrocnemius medialis (GM) and soleus (SOL), subjects were instructed to perform ankle plantar flexion MVCs from a long sitting position with the hip flexed at 90 degrees, while the right foot was secured to a custom-made metal structure and the ankle was maintained in a neutral position. The MVC value for each muscle was defined as the average value around the peak recorded. The %mEMG and %iEMG of each muscle were calculated during the single-leg support phases of six steps, which were also used for time parameter and vertical ground reaction force analyses. The mean %mEMG and %iEMG values across these six steps were compared between 44 bpm and 132 bpm.

## Data analysis

All the data are presented as the mean ± standard deviation. The intra-rater reliability of single-leg support time, time difference and vertical ground reaction force variables across six steps was assessed using the intraclass correlation coefficient (ICC). To ensure that no significant changes occurred in step dynamics—including single-leg support time, time differences, and vertical ground reaction force variables—among the six analysed steps at each tempo, a one-way repeated-measures analysis of variance with Bonferroni-corrected pairwise post hoc comparisons was used. To account for the increased risk of type I error due to multiple comparisons among steps, Bonferroni correction was applied to these within-tempo comparisons.

Before testing for significant differences between the slow and fast tempo conditions, the Shapiro–Wilk test was used to assess the normality of the difference scores for each measurement variable (i.e., single-leg support time, time difference, vertical ground reaction force variables, %mEMG, and %iEMG). If the assumption of normality was not met, the non-parametric Wilcoxon signed-rank test was applied; otherwise, a paired Student's t-test was used. Pearson's correlation was used to assess the relationships between the rate of change in the time variables and the rate of change in the EMG variables from the slow to the fast tempo. The effect size (ES: Cohen's d) was calculated and evaluated as trivial (0–0.19), small (0.20–0.49), medium (0.50–0.79) or large (0.80 and greater) [42]. Statistical significance was set to $p < 0.05$.

## Results

Table 1 shows the single-leg support time, time difference, impact force, force required for step transitions, %mEMG and %iEMG during the stepping test at the slow and fast tempos. No significant differences were found in the temporal variables of step dynamics (single-leg support time and time difference) or the ground reaction force variables (impact force and force required for step transitions) across the analysed steps at each tempo.

The single-leg support time and time difference during stepping at the slow tempo were greater than those at the fast tempo (single-leg support time: $p < 0.01$, ES = 2.591; time difference: $p < 0.01$, ES = 1.362). The ICCs (1,6) for single-leg support time and time difference across six steps were 0.992 and 0.848, respectively, at the slow tempo, and 0.995 and 0.959, respectively, at the fast tempo (n = 12, $p < 0.001$ for all).

The impact force was significantly greater at the fast tempo than at the slow tempo ($p = 0.01$, ES = 0.895). Specifically, the impact force increased by 109.8 ± 11.0% at the fast tempo relative to the slow tempo. In contrast, the force required for step transitions did not differ significantly between the slow- and fast-tempo conditions ($p > 0.05$). The ICCs (1,6) for impact force and the force required for step transitions across six steps were 0.863 and 0.810, respectively, at the slow tempo, and 0.974 and 0.923, respectively, at the fast tempo (n = 12, $p < 0.001$ for all).

The %mEMG of the RF ($p < 0.01$, ES = 0.942), VL ($p < 0.05$, ES = 0.674) and SOL ($p < 0.01$, ES = 1.215) was significantly lower during stepping at the slow tempo than at the fast tempo. The %mEMG of the RF, VL and SOL was lower by 62.6 ± 18.0%, 17.4 ± 29.6% and 25.0 ± 23.2%, respectively, at the slow tempo than at the fast tempo. Conversely, the %iEMG of the VL ($p < 0.01$, ES = 1.300), BF ($p < 0.01$, ES = 0.754), GM ($p < 0.01$, ES = 0.788) and SOL ($p < 0.01$, ES = 2.281) was significantly higher during stepping at the slow tempo than at the fast tempo. The %iEMG values for the VL, BF, GM

**Table 1. Time parameters and lower limb muscle activation during the single-leg support phase in the stepping test at the slow and fast tempo.**

|  |  | Slow (44 bpm) |  |  | Fast (132 bpm) |  |  | p |
|---|---|---|---|---|---|---|---|---|
| Time parameters |  |  |  |  |  |  |  |  |
| Single-leg support time | (s) | 1.02 | ± | 0.27 | 0.31 | ± | 0.07 | * |
| Time difference | (s) | 0.16 | ± | 0.09 | 0.03 | ± | 0.02 | * |
| Vertical ground reaction force parameters |  |  |  |  |  |  |  |  |
| Impact force | (N/kg) | 20.74 | ± | 0.86 | 22.78 | ± | 2.50 | * |
| Required for step transitions force | (N/kg) | 20.04 | ± | 0.50 | 19.56 | ± | 0.99 |  |
| %mEMG |  |  |  |  |  |  |  |  |
| RF | (%) | 7.81 | ± | 5.31 | 24.94 | ± | 22.05 | * |
| VL | (%) | 29.04 | ± | 17.22 | 36.21 | ± | 17.49 | + |
| BF | (%) | 23.01 | ± | 23.15 | 22.00 | ± | 14.81 |  |
| GM | (%) | 77.89 | ± | 55.17 | 101.10 | ± | 64.17 |  |
| SOL | (%) | 78.14 | ± | 23.72 | 109.58 | ± | 39.90 | * |
| %iEMG |  |  |  |  |  |  |  |  |
| RF | (%) | 8.67 | ± | 6.12 | 8.46 | ± | 7.78 |  |
| VL | (%) | 32.02 | ± | 19.73 | 12.24 | ± | 6.31 | * |
| BF | (%) | 25.94 | ± | 27.09 | 7.13 | ± | 4.40 | * |
| GM | (%) | 86.15 | ± | 63.21 | 34.11 | ± | 22.38 | * |
| SOL | (%) | 86.73 | ± | 28.52 | 37.46 | ± | 15.75 | * |

All values are shown as the mean ± standard deviation. Time difference: difference between the metronome sound at each tempo and the subject's right foot contact time; mEMG: mean electromyography signal; RF: rectus femoris; VL: vastus lateralis; BF: biceps femoris; GM: gastrocnemius medialis; SOL: soleus muscle; iEMG: integrated electromyography signal. * and + denote significant differences between the slow and fast tempos at $p \leqq 0.01$ and $p < 0.05$, respectively.

and SOL were 171.3±92.8%, 314.6±361.3%, 214.1±251.0% and 156.8±117.8% higher, respectively, at the slow tempo than at the fast tempo.

There were no significant relationships among the rate of change in the single-leg support time, the time difference from the slow to the fast tempo, or the rate of change in the %mEMG and %iEMG of each muscle.

## Discussion

The aim of this study was to determine whether and which lower limb muscles are required for maintaining dynamic balance in the stepping test by comparing time parameters and muscle activation between slow and fast tempos. As hypothesized, the present results showed that both the single-leg support time and the time difference between foot contact and metronome sound were greater at the slow tempo than at the fast tempo, indicating increased difficulty in maintaining balance at slower tempos. Furthermore, as expected, %iEMG values increased during the more unstable slow tempo condition; however, contrary to our hypothesis, no significant correlations were found between these increases and the time parameters. These findings suggest that while slower tempo stepping prolongs single-leg support time and increases total lower limb muscle activity, dynamic balance indicators are likely influenced by additional factors beyond lower limb muscle activation in young adults.

The results of this study demonstrated that, similar to older adults in a previous study [33], young adults exhibited a significantly greater time difference between foot contact and the metronome beat—a dynamic balance indicator in the stepping test—under the slow tempo condition (44 bpm) than under the fast tempo condition (132 bpm). This time difference is considered a useful balance indicator because, in older adults with reduced lower limb muscle strength, it becomes difficult to maintain stable single-leg support, particularly under the slow tempo condition, where the single-leg support phase is prolonged. This instability disrupts the timing of foot contact, making synchronization with the metronome more challenging [12]. Stepping movements have been reported to be suitable for evaluating dynamic balance ability because they involve displacement of the CoG of the body during movement and instability associated with single-leg support [28]. Moreover, single-leg standing reduces the base of support for the CoG, increasing postural instability and thereby necessitating greater lower limb muscle activity to maintain balance [23]. Indeed, previous studies have shown that in older adults with reduced walking ability and a history of falls [43] or lower knee extension strength [13], the time difference under slow tempo conditions, such as 40 bpm, becomes markedly larger. Furthermore, despite the slow tempo, these individuals exhibit shorter single-leg support times than at fast tempos, indicating difficulty in maintaining stable single-leg support [12]. Therefore, it is considered beneficial to assess single-leg support time alongside the time difference when evaluating dynamic balance in the stepping test. In particular, a large time difference at a slow tempo combined with a short single-leg support time may suggest greater involvement of the lower limb muscles in maintaining balance [13].

In contrast, in the young adult subjects of the present study, although the time difference increased under the slow tempo condition, the single-leg support time did not decrease. Instead, it was approximately three times longer than that in the fast tempo condition, in proportion to the tempo ratio. This suggests that young adults possess sufficient lower limb strength to maintain single-leg support even at slow tempos and may adopt different movement strategies from older adults. At slower tempos, extension movements of the hip, knee, and ankle joints are performed more gradually, and the duration of force generation by the muscles surrounding these joints during the single-leg support phase is extended. Dynamic balance ability in the stepping test is positively correlated with the strength of the knee extensor muscles [13]. Activation of the quadriceps and hamstrings increased during knee joint flexion and extension when laterally raising and lowering the CoG of the body in the stepping task [44]. Consequently, the increase in %iEMG from fast to slow tempos is likely attributable primarily to prolonged movement duration, rather than necessarily reflecting increased muscle activity due to postural instability. In addition, in young subjects, the time difference between foot contact and the metronome sound in the stepping test may not directly represent dynamic balance ability but rather serve as an indirect indicator of rhythm synchronization ability or motor timing accuracy. Humans possess the capacity for sensory-motor synchronization,

defined as the coordinated temporal relationship between body movements and rhythmic patterns in the environment [45]. This ability enables humans not only to react to stimuli but also to anticipate periodic stimuli and synchronize their movements accordingly. Sensory-motor synchronization has been reported to be most effective within a frequency range of approximately 0.5–10 Hz [46]. The metronome tempos used in the present study (44 bpm = 0.73 Hz and 132 bpm = 2.2 Hz) fall within this range, suggesting that subjects may have synchronized their stepping with the auditory stimuli by employing two corrective processes: phase correction (minimizing asynchrony between stimulus and response timing) and period correction (minimizing mismatch in tempo between stimulus and response). Particularly in young adults, who generally have greater muscle strength than older adults, the time difference in stepping may reflect supplementary aspects of timing control and rhythm synchronization, rather than directly representing dynamic balance ability. These factors may explain why no significant association was found between the rate of change in time difference and the rate of change in %iEMG in the present study. Taken together, these findings indicate that further investigation is needed to clarify the extent to which the time difference reflects dynamic balance ability in young adults. To improve the accuracy of dynamic balance assessment in this population, it may be effective to introduce stepping tasks with increased difficulty. For example, stepping that involves forward/backward or lateral movements [33,47], stepping with visual occlusion, or stepping at random rather than constant tempos could impose higher dynamic balance demands than in-place stepping. Such tasks would require greater lower limb muscle force generation, potentially allowing for a clearer evaluation of the relationship between balance ability and muscle activity during stepping. Furthermore, applying different task settings for older and young adults may enable a more detailed comparison of their respective dynamic balance strategies.

One possible explanation for the lack of association between the rate of increase in %iEMG and the increase in the time difference when transitioning from the fast tempo to the slow tempo is that factors beyond lower limb muscle force production—such as vestibular function—may also play a role. The stepping movement coordinated with the stipulated metronome tempo cannot rely solely on the performer's intrinsic stable timing. During stepping, the CoG of the body must move vertically and mediolaterally while the stance leg remains stationary as the supporting limb. Dynamic balance during regularly repeated movements, such as stepping, is mechanically lost and reset [48]. Human balance ability is associated with the vestibular, visual, somatosensory, and musculoskeletal systems [14]. Maintaining dynamic balance during the stepping test requires the ability to make foot contact in accordance with sound, and vestibular postural control may have more direct importance than muscular strength. In addition, human postural control involves regulated ankle-joint strategies (anteroposterior stability) and hip-joint strategies (mediolateral stability) [19]. Mediolateral ankle stability may involve the tibialis anterior, peroneus longus, peroneus brevis, and gastrocnemius lateralis muscles [49]. Beyond these muscles, dynamic balance is also considered a measure of core stability. For example, during the star excursion balance test, trunk muscles such as the erector spinae, external oblique, and rectus abdominis are functionally recruited depending on the reach direction, and play an important role in single-leg reaching tasks [50]. Similarly, during the Y-balance test on unstable surfaces, high activation levels of trunk muscles such as the external oblique and erector spinae have been reported as being necessary to maintain a stable single-leg posture [39]. Taken together, these findings suggest that in young adults, the ability to maintain balance during stepping may be influenced not only by the force-generating capacity of specific lower limb muscles, but also by other balance-related functions, such as sensory integration, and by the contribution of lower limb or trunk muscles not measured in the present study. However, because the present study did not record activity from these muscles, we cannot address their potential contribution to the time-difference parameter, which served as an indicator of dynamic stability during stepping.

The present study found that mean EMG (%mEMG) values of the lower limb muscles (RF, VL, SOL) were greater at the fast tempo than at the slow tempo, whereas integrated EMG (%iEMG) values (VL, BF, GM, SOL) were greater at the slow tempo. This pattern indicates that the evaluation of muscle activity during stepping is method-dependent and reflects the specific influence of movement tempo. The mEMG and iEMG represent distinct physiological aspects of muscle activation: mEMG reflects the average intensity of muscle activation over a given period, whereas iEMG reflects the

cumulative muscle activity over time [27]. The impact force at foot contact was significantly greater in fast-tempo stepping than in slow-tempo stepping, likely resulting from the foot contacting the ground from a greater vertical displacement, which increases deceleration forces upon landing. During this instant, the lower limb muscles must rapidly generate force to absorb the impact and stabilize body posture [51]. Additionally, fast-tempo stepping demands greater instantaneous force generation during the transition phase when shifting the CoG of the body, contributing to higher mEMG values. These findings align with the notion that mEMG captures short-term, high-intensity bursts of muscle activation, whereas iEMG reflects sustained activation over longer periods. Conversely, because iEMG incorporates the temporal component of muscle activation, it is sensitive to prolonged contraction durations and thus likely to increase under slow-tempo conditions due to the extended single-leg support phase [52,53]. This sustained phase requires continuous force generation to maintain postural stability, which is captured by higher iEMG values. Taken together, the results suggest that lower limb muscles in the stepping test serve dual functions: (1) generating rapid force to counteract impact and facilitate CoG shifts—more prominent in fast-tempo stepping—and (2) sustaining force over prolonged single-leg support phases—more prominent in slow-tempo stepping. This dual demand may partly explain the lack of a direct relationship between the rate of change in stability indices and the rate of change in lower limb muscle activity. From a functional and clinical application perspective, when using the stepping test as part of a rehabilitation or fitness exercise program, it may be desirable to tailor the stepping tempo to the individual's goals. For example, if the purpose is to enhance the ability for instantaneous force generation, fast-tempo stepping may be preferable, whereas if the aim is to improve sustained force generation while considering injury prevention, slow-tempo stepping may be more appropriate.

In this study, large standard deviations were observed in the temporal variables and muscle activity measurements during the stepping test at each tempo. This result may be attributed to variations in the stepping movements among the subjects. Regarding the relationship between joint angles and muscle activity, the magnitude of muscle activity is known to be influenced by joint angles [54]. Differences in stepping movements between tempos may be related to landing mechanics associated with ankle plantarflexion and dorsiflexion angles. Although forefoot landing was confirmed at both tempos during practice, at a faster tempo the landing position may have shifted further forward compared with a slower tempo. In running, increased speed has been reported to be associated with greater ankle plantarflexion [55] or a transition from rearfoot to forefoot landing [56]. Foot strike patterns are modulated by neuromuscular control mechanisms, with adjustments in the timing, duration, and peak activation of muscle synergies playing a crucial role [57]. Indeed, variations in landing mechanics influence lower limb muscle activity, as forefoot landing has been shown to increase posterior lower leg muscle activity compared to rearfoot landing [58]. Furthermore, differences in landing patterns, such as forefoot or rearfoot landings, may have impacted lower limb muscle activity [59]. Therefore, in the stepping test, adaptations in landing mechanics or ankle joint movements in response to movement speed may have influenced lower limb muscle activity, potentially contributing to the greater standard deviation observed in muscle activity levels. In addition to landing mechanics, hip joint flexion and extension movements resulting from vertical thigh motion may also be involved. During the stepping test, the subjects were instructed to land in synchronization with the metronome sound to the best of their ability. However, this instruction may have resulted in variations in the flexion angles of the hip and knee joints, resulting in differences in the elevation heights of the thigh and lower leg. These differences in the downwards swing height of the lower limb may have affected muscle activity during single-leg support and the timing of landing. Similarly, the choice of landing on the forefoot or rearfoot likely influences lower limb muscle activity. Therefore, interindividual differences in hip, knee, and ankle joint movements, as well as landing patterns during stepping, may have contributed to the large standard deviations observed in the measured variables. Considering these factors, it may be necessary to combine EMG with motion analysis using a camera system when evaluating dynamic balance indices during stepping in young adults. Furthermore, the lower sampling frequency for ground reaction force and muscle activity in this study compared with a previous study [49] may have introduced additional measurement errors.

This study demonstrated that the tempo of the stepping test significantly affects dynamic balance stability and lower limb muscle activity in young men of the same age group with a BMI of less than 25. However, several limitations should be acknowledged. First, only 12 subjects were included in the correlation analysis conducted, representing a relatively small sample size. Future studies with larger cohorts are necessary to identify specific muscles involved in dynamic stability during the stepping test. Stepping is a movement in which the CoG of the body alternately shifts from side to side, and the ability to reverse this movement may vary depending on tempo, potentially affecting lower limb muscle activity. In the present study, following previous research, the time difference between foot contact and the metronome sound was used as the index of dynamic balance. However, whether dynamic balance can be fully evaluated using time difference alone remains uncertain. In young adults, the time difference may not directly represent dynamic balance ability, but rather serve as an indirect indicator of rhythm synchronization ability or motor timing accuracy. Future studies should consider incorporating additional measures, such as centre of pressure trajectory, velocity, and area. In light of this, although only the vertical component of the ground reaction force was measured in this study, the mediolateral component should also be assessed. Furthermore, dynamic stability depends not only on the peroneal muscles and the tibialis anterior [49], but also on the trunk muscles [60,61]. As this study did not record EMG activity from these muscles, the relative contributions of trunk and lower limb muscles to dynamic stability during the stepping test could not be determined. In addition, the subjects in this study were young adults, whereas the clinical relevance of stepping-based dynamic balance assessments is often greater in older adults. Aging is associated with shorter single-leg support time [12], increased reliance on proximal segments (hip/trunk) [62], and changes in balance strategies. Therefore, the present findings may not be generalizable to older populations. Future studies should include older adults to clarify age-related differences in balance control mechanisms during stepping. Finally, the stepping motion itself may have influenced the measured variables. Future analyses should take into account factors such as hip, knee, and ankle joint angles, as well as landing patterns.

In the stepping test under the slow-tempo condition, the maintenance time of the base of support was prolonged, requiring sustained force generation by the lower limb muscles. As reported in previous studies, the time difference between the metronome sound and foot contact, which has been considered an indicator of dynamic balance ability, also increased under this condition [12]. However, it remains unclear whether the magnitude of this time difference directly reflects dynamic balance ability in young adults. At present, it may be more appropriate to interpret this measure not as a direct indicator of balance ability, but as an indirect indicator of sensory-motor synchronization—that is, rhythm synchronization or timing control ability [45]. Considering this, the stepping test may have greater functional and clinical significance for older adults with reduced lower limb strength, serving as a training method to improve dynamic balance ability or as a safe assessment tool. In contrast, in young adults, the test may have greater functional and clinical relevance for assessing timing or rhythmic abilities. Therefore, caution should be exercised when interpreting whether this measure directly reflects dynamic balance ability in young adults.

## Conclusions

The aim of this study was to determine whether and which lower limb muscles are required for maintaining dynamic balance during the stepping test. This study demonstrated that, in young adults, slow-tempo stepping required sustained force generation by the lower limb muscles, whereas fast-tempo stepping required rapid, explosive force generation, and no single muscle was consistently associated with the dynamic balance indicators during stepping.

## Acknowledgments

The authors thank all the subjects who participated in this study.

## Author contributions

**Conceptualization:** Keiji Koyama.

**Data curation:** Keiji Koyama, Yusuke Oyama.

**Formal analysis:** Keiji Koyama, Chisa Watanabe, Yusuke Oyama.

**Funding acquisition:** Keiji Koyama.

**Investigation:** Keiji Koyama, Chisa Watanabe, Yusuke Oyama.

**Methodology:** Keiji Koyama, Chisa Watanabe, Yusuke Oyama.

**Project administration:** Keiji Koyama, Chisa Watanabe, Yusuke Oyama.

**Resources:** Keiji Koyama.

**Software:** Keiji Koyama.

**Supervision:** Keiji Koyama.

**Validation:** Keiji Koyama.

**Visualization:** Keiji Koyama.

**Writing – original draft:** Keiji Koyama.

**Writing – review & editing:** Keiji Koyama, Yusuke Oyama.

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
