## [Decision Letter · Decision Letter 0]

2 Aug 2025

Dear Dr. Koyama,

Thank you for submitting your manuscript to PLOS ONE. After careful consideration, we feel that it has merit but does not fully meet PLOS ONE’s publication criteria as it currently stands. Therefore, we invite you to submit a revised version of the manuscript that addresses the points raised during the review process.

**ACADEMIC EDITOR:**plosone@plos.org . A rebuttal letter that responds to each point raised by the academic editor and reviewer(s). You should upload this letter as a separate file labeled 'Response to Reviewers'.A marked-up copy of your manuscript that highlights changes made to the original version. You should upload this as a separate file labeled 'Revised Manuscript with Track Changes'.An unmarked version of your revised paper without tracked changes. You should upload this as a separate file labeled 'Manuscript'.

We look forward to receiving your revised manuscript.

Kind regards,

Emiliano Cè, Ph.D.

Academic Editor

PLOS ONE

Journal Requirements:

K.K. (Keiji Koyama) was supported by a Grant-in-Aid for Scientific Research from the Japan Society for the Promotion of Science (Grant-in-Aid for Young Scientists (19K20021) and Grant-in-Aid for Scientific Research (23K10674)).

The authors declare that they have no known competing financial or personal interests that could have appeared to influence the work reported in this paper.

4. In the online submission form, you indicated that the data that support the findings of this study are not publicly available due to ethical restrictions, but are available from the corresponding author upon reasonable request.

Reviewers' comments:

Reviewer's Responses to Questions

**Comments to the Author**

1. Is the manuscript technically sound, and do the data support the conclusions?

Reviewer #1: Yes

Reviewer #2: Partly

2. Has the statistical analysis been performed appropriately and rigorously?

Reviewer #1: Yes

Reviewer #2: Yes

3. Have the authors made all data underlying the findings in their manuscript fully available?

Reviewer #1: Yes

Reviewer #2: No

4. Is the manuscript presented in an intelligible fashion and written in standard English?

Reviewer #1: Yes

Reviewer #2: Yes

Reviewer #1: General Comments:

Overall, this is a good manuscript that presents how stepping tempo affects lower limb muscle activation and dynamic balance in young men, adding useful insight for rehabilitation and training contexts. The study design is methodologically solid, with a clear rationale and appropriate EMG and force plate measures. The introduction reviews the relevant biomechanical literature well but could frame a sharper hypothesis. The methods are described in sufficient detail for replication, and the results are clearly presented with appropriate stats and effect sizes. The discussion interprets the biomechanical implications accurately, comparing them to relevant previous studies (e.g., Shin & Demura, Lux et al.), but would benefit from deeper linkage to comparable EMG literature and clearer acknowledgment of limitations. Overall, the paper is well-structured, but there is room for improvement, specifically with the discussion.

Specific Comments:

• Abstract: Consider including specific results or quantitative outcomes in the abstract. Currently it states the general findings, but adding some numbers (e.g., percentage reductions in EMG, etc) make it more clear.

L45–109 (Introduction): The introduction is well-written and references relevant stepping, balance, and EMG studies (e.g., Shin & Demura). However, it does not explicitly state a hypothesis. Recommend adding a sentence to clarify the expectation: e.g., "It was hypothesized that stepping at slower tempos would increase integrated muscle activation due to prolonged single-leg stance."

L163–180 (Experimental Procedure): The force plate setup is described clearly, but specify if two force plates were used simultaneously for bilateral data or if only the right leg was analyzed. Similar studies (Lux et al., 1992) analyzed bilateral data.

L182–194 (Time Parameters): Only the right leg’s single-leg support is analyzed — explain why you chose the right leg only. If you assume limb symmetry, justify it.

L204–222 (Vertical GRF): the filter description is awkwardly worded. Consider rewriting for clarity: “Signals were low-pass filtered at 20 Hz using a zero-phase-lag FIR filter with a Kaiser window…” Also, clarify if vertical only or other force components were measured — as mediolateral forces could affect balance.

L232–279 (EMG Analysis): clarify whether the electrode placement followed SENIAM guidelines. Also, indicate whether crosstalk between adjacent muscles was considered.

L281–296 (Data Analysis): It would strengthen the text to mention whether you controlled for multiple comparisons formally. Also, you might clarify whether assumptions for parametric tests were checked.

L298–337 (Results): The results are clear and effect sizes provided. The large standard deviations for EMG suggest high inter-individual variability — discuss this more in the Discussion (which you partly do).

• L343–354 (Discussion): The main finding is correctly interpreted: slow tempo increases balance demand and integrated EMG. Connect this more directly to studies like Shin & Demura (2007, 2009) and Lux et al. (1992) who found greater quadriceps activation in step-up exercises. This strengthens your comparison to prior EMG literature.

• L355–394: you mention that muscles like the tibialis anterior and peroneals may play a role — you should note that these muscles were not recorded. Recommend clarifying this as a limitation.

• L411–434: The explanation of impact force differences is logical, but it would help to connect this to landing mechanics literature (e.g., foot strike patterns in running). You do this later — consider merging or cross-referencing these points.

• L437–451: You acknowledge large SDs and inter-individual variability well. Consider suggesting that a motion capture analysis could confirm joint angle influences.

• L455–468 (Limitations): add that you did not record trunk EMG, which could contribute to mediolateral stability, as prior studies suggest (e.g., Kaur et al., 2022). Also, your subjects are young adults, but the study's findings are more applicable to the older population. You should clarify and explain this extensively in the limitations.

Reviewer #2: General comment

This study addresses an important question regarding which lower limb muscles contribute to dynamic balance during a stepping test. However, the main conclusion – that “maintaining dynamic balance requires force generation from all lower limb muscles rather than a specific one” – requires more cautious interpretation based on the data presented. I recommend considering the following points for revision.

#Major comments

1. Interpretation of iEMG values

iEMG is strongly influenced by movement duration, which may partly explain the larger values observed at the slow tempo. It seems self-evident that iEMG would be greater in the slow condition due to the longer step duration. Therefore, please clarify the rationale for using iEMG and explain what new insights were gained from comparing it with mEMG.

2. Validity of using metronome synchronization error as a balance index

If the metronome tempo is constant, participants may be able to maintain a steady stepping rhythm even without auditory cues. Thus, the time difference between foot contact and the metronome sound may not fully reflect dynamic balance ability. Although you referenced previous studies for using this metric, please provide additional justification for its validity in this context. For instance, would introducing random tempo variations improve the reliability of this measure as a balance index?

3. mEMG findings contrary to the hypothesis

The mEMG values were higher in the fast tempo condition, which contradicts the hypothesis that muscle activity would increase under the more unstable slow tempo condition. Please provide additional discussion on why this might have occurred.

4. Interpretation of single-leg support time

The longer single-leg support time at the slow tempo seems like an expected outcome due to slower stepping. Please clarify the significance of presenting and statistically testing this result. How does this difference meaningfully contribute to evaluating dynamic balance ability?

5. Practical significance of time difference

Although the time difference between foot contact and metronome sound was statistically different between tempos, its functional or clinical relevance is unclear. Please discuss the practical implications of this difference.

6. Statistical analysis description

The manuscript states that “measurement variables (single-leg support time, time difference, ground reaction force variables, %mEMG, and %iEMG) were used as dependent variables, and tempo (slow/fast) as an independent variable for paired t-tests.” This description is inappropriate because paired t-tests do not use the framework of dependent and independent variables. Please revise this statement for accuracy.

#Minor comments

7. Figure 1 resolution and labeling

The resolution of Figure 1 is low, making it difficult to read. Additionally, the figure lacks legends for the rectangular and continuous signals. I assume the rectangular signal indicates metronome timing, but is its width precisely aligned with the metronome interval? If the continuous signal represents ground reaction force (with a 10 N threshold), please indicate the exact points of foot contact and toe-off in the figure.

8. Raw vs. processed data in Figure 2

Figure 2 seems to illustrate data processing for ground reaction forces. It would improve transparency if you also provide raw and processed signals for both ground reaction forces and EMG to allow readers to evaluate the effect of filtering.

**Do you want your identity to be public for this peer review?** For information about this choice, including consent withdrawal, please see our Privacy Policy

Reviewer #1: No

Reviewer #2: No

---

## [Author Response · Author response to Decision Letter 1]

2 Sep 2025

We sincerely appreciate the reviewers for their careful and thorough evaluation of our manuscript. Revising it in light of their critiques has been extremely beneficial, and we are confident that their comments have greatly improved both the content and overall quality of our work. Wherever possible, we have incorporated the reviewers’ criticisms, questions, and suggestions to enhance the quality of the manuscript.

Our responses to the comments are presented in the order in which they were raised. Many of our replies required insertions and modifications within the manuscript itself, which are indicated in the revised annotated version with (Page, Line) notation. All changes in the revised manuscript are highlighted, and a list of the revisions is provided together with our point-by-point responses to the reviewers’ comments.

We have also included in this cover letter our confirmations and responses to the Journal Requirements, as well as our replies to both reviewers regarding the revisions. We kindly ask you to review these materials.

------------

Journal Requirements:

→I have reviewed the instructions and revised the manuscript to comply with the PLOS ONE style requirements.

K.K. (Keiji Koyama) was supported by a Grant-in-Aid for Scientific Research from the Japan Society for the Promotion of Science (Grant-in-Aid for Young Scientists (19K20021) and Grant-in-Aid for Scientific Research (23K10674)). Please state what role the funders took in the study. If the funders had no role, please state: "The funders had no role in study design, data collection and analysis, decision to publish, or preparation of the manuscript."

If this statement is not correct you must amend it as needed. Please include this amended Role of Funder statement in your cover letter; we will change the online submission form on your behalf. 「The funders had no role in study design, data collection and analysis, decision to publish, or preparation of the manuscript.」

→The funders were not involved in the study design, data collection and analysis, decision to publish, or preparation of the manuscript. Therefore, the statement “The funders had no role in study design, data collection and analysis, decision to publish, or preparation of the manuscript. ” is correct. I would appreciate it if you could make this change. I would be grateful if you could update the online submission form accordingly.

The authors declare that they have no known competing financial or personal interests that could have appeared to influence the work reported in this paper. Please confirm that this does not alter your adherence to all PLOS ONE policies on sharing data and materials, by including the following statement: "This does not alter our adherence to PLOS ONE policies on sharing data and materials.” (as detailed online in our guide for authors http://journals.plos.org/plosone/s/competing-interests). If there are restrictions on sharing of data and/or materials, please state these. Please note that we cannot proceed with consideration of your article until this information has been declared. Please include your updated Competing Interests statement in your cover letter; we will change the online submission form on your behalf.

→I have confirmed this. Please add the following statement to the Competing Interests section: "This does not alter our adherence to PLOS ONE policies on sharing data and materials.” I would be grateful if you could update the online submission form accordingly.

4. In the online submission form, you indicated that the data that support the findings of this study are not publicly available due to ethical restrictions, but are available from the corresponding author upon reasonable request. All PLOS journals now require all data underlying the findings described in their manuscript to be freely available to other researchers, either a. In a public repository, b. Within the manuscript itself, or c. Uploaded as supplementary information. This policy applies to all data except where public deposition would breach compliance with the protocol approved by your research ethics board. If your data cannot be made publicly available for ethical or legal reasons (e.g., public availability would compromise patient privacy), please explain your reasons on resubmission and your exemption request will be escalated for approval.

→I have reviewed the requirement. All data underlying the findings of this study are already provided within the manuscript, and thus comply with the PLOS ONE data availability policy. I have also updated the information in the online submission form when submitting the revised manuscript, and I would appreciate it if the editorial office could kindly confirm this on your end as well.

→Thank you for your suggestion. I have carefully evaluated the recommended references and revised the manuscript as appropriate

---------------

Review Comments to the Author

Reviewer #1: General Comments:

Overall, this is a good manuscript that presents how stepping tempo affects lower limb muscle activation and dynamic balance in young men, adding useful insight for rehabilitation and training contexts. The study design is methodologically solid, with a clear rationale and appropriate EMG and force plate measures. The introduction reviews the relevant biomechanical literature well but could frame a sharper hypothesis. The methods are described in sufficient detail for replication, and the results are clearly presented with appropriate stats and effect sizes. The discussion interprets the biomechanical implications accurately, comparing them to relevant previous studies (e.g., Shin & Demura, Lux et al.), but would benefit from deeper linkage to comparable EMG literature and clearer acknowledgment of limitations. Overall, the paper is well-structured, but there is room for improvement, specifically with the discussion.

→Thank you for your valuable comments. Based on your suggestions, I have revised the manuscript, particularly the Discussion section. I would appreciate it if you could review the revised version.

Specific Comments:

Abstract: Consider including specific results or quantitative outcomes in the abstract. Currently it states the general findings, but adding some numbers (e.g., percentage reductions in EMG, etc) make it more clear.

→Retouched >>>(Page 2, Line 23-26)

L45–109 (Introduction): The introduction is well-written and references relevant stepping, balance, and EMG studies (e.g., Shin & Demura). However, it does not explicitly state a hypothesis. Recommend adding a sentence to clarify the expectation: e.g., "It was hypothesized that stepping at slower tempos would increase integrated muscle activation due to prolonged single-leg stance."

→Retouched >>>(Page 4, Line 96 to Page 5, Line 108)

L163–180 (Experimental Procedure): The force plate setup is described clearly, but specify if two force plates were used simultaneously for bilateral data or if only the right leg was analyzed. Similar studies (Lux et al., 1992) analyzed bilateral data.

→Retouched >>>(Page 7, Line 168 and Line 175-179)

L182–194 (Time Parameters): Only the right leg’s single-leg support is analyzed — explain why you chose the right leg only. If you assume limb symmetry, justify it.

→Retouched >>>(Page 7, Line 185-189)

L204–222 (Vertical GRF): the filter description is awkwardly worded. Consider rewriting for clarity: “Signals were low-pass filtered at 20 Hz using a zero-phase-lag FIR filter with a Kaiser window…”

→Retouched >>>(Page 9, Line 229)

Also, clarify if vertical only or other force components were measured

→Retouched >>>(Page 8, Line 220)

— as mediolateral forces could affect balance.

→(Page 19, Line 550-553)

L232–279 (EMG Analysis): clarify whether the electrode placement followed SENIAM guidelines.

→Retouched >>>(Page 10, Line 265)

Also, indicate whether crosstalk between adjacent muscles was considered.

→Retouched >>>(Page 10, Line 272)

L281–296 (Data Analysis): It would strengthen the text to mention whether you controlled for multiple comparisons formally.

→Retouched >>>(Page 11, Line 310-316)

Also, you might clarify whether assumptions for parametric tests were checked.

→Retouched >>>(Page 12, Line 318-323)

L298–337 (Results): The results are clear and effect sizes provided. The large standard deviations for EMG suggest high inter-individual variability — discuss this more in the Discussion (which you partly do).

→Added and retouched>>>(Page 18, Line 505-523)

L343–354 (Discussion): The main finding is correctly interpreted: slow tempo increases balance demand and integrated EMG. Connect this more directly to studies like Shin & Demura (2007, 2009) and Lux et al. (1992) who found greater quadriceps activation in step-up exercises. This strengthens your comparison to prior EMG literature.

→Added and retouched>>>(Page 14, Line 412-425)

L355–394: you mention that muscles like the tibialis anterior and peroneals may play a role — you should note that these muscles were not recorded. Recommend clarifying this as a limitation.

→Added>>>(Page 17, Line 461-468)

→Added>>>(Page 20, Line 553-557)

L411–434: The explanation of impact force differences is logical, but it would help to connect this to landing mechanics literature (e.g., foot strike patterns in running). You do this later — consider merging or cross-referencing these points.

→Retouched>>>(Page 18, Line 505-Page 19, Line 523)

L437–451: You acknowledge large SDs and inter-individual variability well. Consider suggesting that a motion capture analysis could confirm joint angle influences.

→Added>>>(Page 19, Line 533-535)

L455–468 (Limitations): add that you did not record trunk EMG, which could contribute to mediolateral stability, as prior studies suggest (e.g., Kaur et al., 2022).

→Added>>>(Page 16, Line 455-461)

→Added>>>(Page 20, Line 553-557)

Also, your subjects are young adults, but the study's findings are more applicable to the older population. You should clarify and explain this extensively in the limitations.

→Added>>>(Page 20, Line 557-563)

Reviewer #2: General comment

This study addresses an important question regarding which lower limb muscles contribute to dynamic balance during a stepping test. However, the main conclusion – that “maintaining dynamic balance requires force generation from all lower limb muscles rather than a specific one” – requires more cautious interpretation based on the data presented. I recommend considering the following points for revision.

→Thank you very much for your valuable comment. As you pointed out, it is difficult to determine from our data whether the maintenance of dynamic balance during stepping in young adults is attributable to force generation from all lower limb muscles or to the prolonged single-leg support time. Therefore, we have revised the discussion and conclusion to provide a more cautious interpretation based on the results. We kindly ask you to review the revised version.

#Major comments

1. Interpretation of iEMG values

iEMG is strongly influenced by movement duration, which may partly explain the larger values observed at the slow tempo. It seems self-evident that iEMG would be greater in the slow condition due to the longer step duration. Therefore, please clarify the rationale for using iEMG

→Added>>>(Page 4, Line 96-Page 5, Line 108)

and explain what new insights were gained from comparing it with mEMG.

→Added>>>(Page 17, Line 470-Page 18, Line 501)

2. Validity of using metronome synchronization error as a balance index

If the metronome tempo is constant, participants may be able to maintain a steady stepping rhythm even without auditory cues. Thus, the time difference between foot contact and the metronome sound may not fully reflect dynamic balance ability. Although you referenced previous studies for using this metric, please provide additional justification for its validity in this context. For instance, would introducing random tempo variations improve the reliability of this measure as a balance index?

→Retouched>>>(Page 14, Line 392-Page 15, Line 410)

→Added>>>(Page 15, Line 412-Page 16, Line 438)

3. mEMG findings contrary to the hypothesis

The mEMG values were higher in the fast tempo condition, which contradicts the hypothesis that muscle activity would increase under the more unstable slow tempo condition. Please provide additional discussion on why this might have occurred.

→Added>>>(Page 17, Line 470-Page 18, Line 501)

4. Interpretation of single-leg support time

The longer single-leg support time at the slow tempo seems like an expected outcome due to slower stepping. Please clarify the significance of presenting and statistically testing this result. How does this difference meaningfully contribute to evaluating dynamic balance ability?

→Added>>>(Page 15, Line 401-410)

→Added>>>(Page 15, Line 412-417)

5. Practical significance of time difference

Although the time difference between foot contact and metronome sound was statistically different between tempos, its functional or clinical relevance is unclear. Please discuss the practical implications of this difference.

→Added>>>(Page 14, Line 391-Page 15, Line 410)

→Added>>>(Page 15, Line 427-Page 16, Line 438)

→Added>>>(Page 19, Line 544-549)

6. Statistical analysis description

The manuscript states that “measurement variables (single-leg support time, time difference, ground reaction force variables, %mEMG, and %iEMG) were used as dependent variables, and tempo (slow/fast) as an independent variable for paired t-tests.” This description is inappropriate because paired t-tests do not use the framework of dependent and independent variables. Please revise this statement for accuracy.

→Retouched>>>(Page 12, Line 318-323)

#Minor comments

7. Figure 1 resolution and labeling

The resolution of Figure 1 is low, making it difficult to read. Additionally, the figure lacks legends for the rectangular and continuous signals. I assume the rectangular signal indicates metronome timing, but is its width precisely aligned with the metronome interval? If the continuous signal represents ground reaction force (with a 10 N threshold), please indicate the exact points of foot contact and toe-off in the figure.

→Changed Figure 1

→Retouched>>>(Page 8, Line 212-217)

8. Raw vs. processed data in Figure 2

Figure 2 seems to illustrate data processing for ground reaction forces. It would improve transparency if you also provide raw and processed signals for both ground reaction forces and EMG to allow readers to evaluate the effect of filtering.

→Changed Figure 2

→Retouched>>>(Page 9, Line 247-250)

---------------

---

## [Decision Letter · Decision Letter 1]

1 Oct 2025

Dear Dr. Koyama,

Thank you for submitting your manuscript to PLOS ONE. After careful consideration, we feel that it has merit but does not fully meet PLOS ONE’s publication criteria as it currently stands. Therefore, we invite you to submit a revised version of the manuscript that addresses the points raised during the review process.

**ACADEMIC EDITOR: **

We look forward to receiving your revised manuscript.

Kind regards,

Emiliano Cè, Ph.D.

Academic Editor

PLOS ONE

Journal Requirements:

Reviewers' comments:

Reviewer's Responses to Questions

**Comments to the Author**

Reviewer #1: All comments have been addressed

Reviewer #2: (No Response)

2. Is the manuscript technically sound, and do the data support the conclusions?

Reviewer #1: Yes

Reviewer #2: Yes

3. Has the statistical analysis been performed appropriately and rigorously?

Reviewer #1: Yes

Reviewer #2: Yes

4. Have the authors made all data underlying the findings in their manuscript fully available?

Reviewer #1: Yes

Reviewer #2: Yes

5. Is the manuscript presented in an intelligible fashion and written in standard English?

Reviewer #1: Yes

Reviewer #2: Yes

Reviewer #1: (No Response)

Reviewer #2: Overall, I find that the revised manuscript has addressed several of the reviewers’ concerns and has improved in clarity and interpretation. I acknowledge these efforts and see value in the main contribution of the study. However, because the quality of the initial submission was not particularly strong, I still have some reservations about whether all of the finer details and interpretations are fully sound. I recommend that the editor take this context into account when deciding on the final disposition of the manuscript.

#Interpretation of iEMG

Regarding the interpretation of iEMG, the authors have clearly distinguished it from mEMG and added discussion on the significance of capturing cumulative muscle activity under the slow-tempo condition, where single-leg support time is prolonged. This explanation clarifies the rationale for adopting iEMG as an analysis variable, and I find this point acceptable.

#Validity of the metronome synchronization error & clinical significance of the time difference

In this study, the time difference between metronome sound and foot contact is used as an indicator of “dynamic balance.” However, I have concerns about presenting this measure as one of the main results. The reason is that the chosen tempos (44 bpm and 132 bpm) are relatively fast, and especially at 132 bpm, movement speed and reaction time constraints are likely to strongly influence timing control. Thus, caution is warranted before concluding that this synchronization error directly reflects dynamic balance. Rather, it may be more appropriate to interpret it as an index of rhythm synchronization ability or motor timing accuracy.

Furthermore, although the difference was statistically significant between tempos, the functional and clinical implications of the magnitude of this difference remain unclear. At present, I believe it is more appropriate to regard it as an auxiliary timing index that may indirectly relate to balance, rather than as a primary balance indicator. Since the main objective of this study is to examine changes in lower limb muscle activity under different tempo conditions, the synchronization error analysis should be positioned as a supplementary result, with its limitations clearly stated. I leave it to the editor to decide whether and how the authors should further address this issue or whether the manuscript is acceptable in its current form.

#mEMG results contrary to the hypothesis

For the finding that mEMG values increased under the fast-tempo condition, the authors attributed this to greater landing impact forces and the need for rapid force generation associated with center-of-gravity shifts. I reviewed the cited references (49, 27, 50, 51) and found that references 49 and 51 provide particularly strong support, while reference 27 supports methodological validity and reference 50 contributes to the discussion of clinical relevance. Therefore, I consider the revised explanation of the mEMG findings to be sufficiently reinforced.

#Interpretation of single-leg support time

Although the longer single-leg support time under the slow-tempo condition is an expected outcome due to slower movement, I agree with the authors that statistically confirming and quantifying this difference has value. Thus, I find the revision on this point acceptable.

#Revision of statistical analysis description

The description of the statistical analyses has been corrected from its previously inaccurate wording and improved to be more appropriate and precise. I am satisfied with this revision.

#Figure 1 (resolution and labeling)

For Figure 1, the labeling has improved; however, the resolution itself does not appear to have been sufficiently enhanced. I would leave it to the editorial office to judge whether the resolution is adequate for publication.

#Figure 2 (raw and processed data presentation)

For Figure 2, the inclusion of both raw and processed data makes the data processing procedure clearer and more transparent. I consider this an improvement.

**Do you want your identity to be public for this peer review?** For information about this choice, including consent withdrawal, please see our Privacy Policy

Reviewer #1: No

Reviewer #2: No

---

## [Author Response · Author response to Decision Letter 2]

6 Oct 2025

Dear reviewers,

Following the first revision, we would like to express our sincere gratitude once again for your careful and thorough evaluation of our manuscript. Revising the paper in light of your comments has again been extremely valuable, and we are confident that your feedback has further improved both the content and the overall quality of the manuscript. We have carefully reviewed all of your critiques, questions, and suggestions, and have incorporated them wherever possible to enhance the completeness and clarity of the paper.

Our responses are presented in the order in which the comments were provided. Many of our replies correspond to insertions or revisions made directly in the manuscript, with the relevant locations indicated by page and line numbers in the newly annotated version. All modifications are highlighted in the revised manuscript, and a detailed list of the changes, together with our point-by-point responses to each reviewer’s comments, is included. In addition, this cover letter summarizes the main revisions made in response to the reviewers’ feedback. We kindly ask you to review these materials.

We hope that the revised version of the manuscript now meets the requirements for publication in PLOS ONE.

Thank you very much for your continued time and consideration.

Sincerely,

------------

Reviewer #2:

We sincerely appreciate your continued careful review of our manuscript and your insightful comments. In particular, your remarks on “#Validity of the metronome synchronization error & clinical significance of the time difference” were extremely valuable and prompted us to reconsider part of our interpretation regarding the time difference results. We are truly grateful for your constructive feedback. The specific revisions and their corresponding locations in the revised manuscript are provided below for your reference.

#Validity of the metronome synchronization error & clinical significance of the time difference

In this study, the time difference between metronome sound and foot contact is used as an indicator of “dynamic balance.” However, I have concerns about presenting this measure as one of the main results. The reason is that the chosen tempos (44 bpm and 132 bpm) are relatively fast, and especially at 132 bpm, movement speed and reaction time constraints are likely to strongly influence timing control. Thus, caution is warranted before concluding that this synchronization error directly reflects dynamic balance. Rather, it may be more appropriate to interpret it as an index of rhythm synchronization ability or motor timing accuracy.

→Added >>>(Page 15, Lines 425–442)

Furthermore, although the difference was statistically significant between tempos, the functional and clinical implications of the magnitude of this difference remain unclear. At present, I believe it is more appropriate to regard it as an auxiliary timing index that may indirectly relate to balance, rather than as a primary balance indicator. Since the main objective of this study is to examine changes in lower limb muscle activity under different tempo conditions, the synchronization error analysis should be positioned as a supplementary result, with its limitations clearly stated. I leave it to the editor to decide whether and how the authors should further address this issue or whether the manuscript is acceptable in its current form.

→Added >>>(Page 20, Lines 566–568)

→Changed and modified >>>(Page 21, Lines 586–600)

#Figure 1 (resolution and labeling)

For Figure 1, the labeling has improved; however, the resolution itself does not appear to have been sufficiently enhanced. I would leave it to the editorial office to judge whether the resolution is adequate for publication.

→Improved the resolution of Figure1 and Figure 2.

------------

---

## [Decision Letter · Decision Letter 2]

30 Oct 2025

Dear Dr. Koyama,

Thank you for submitting your manuscript to PLOS ONE. After careful consideration, we feel that it has merit but does not fully meet PLOS ONE’s publication criteria as it currently stands. Therefore, we invite you to submit a revised version of the manuscript that addresses the points raised during the review process.

**Dear Authors, ****your manuscript was re-evaluated by two experts in the field that reported some minor issues you should consider during the revision process. **plosone@plos.org . A rebuttal letter that responds to each point raised by the academic editor and reviewer(s). You should upload this letter as a separate file labeled 'Response to Reviewers'.A marked-up copy of your manuscript that highlights changes made to the original version. You should upload this as a separate file labeled 'Revised Manuscript with Track Changes'.An unmarked version of your revised paper without tracked changes. You should upload this as a separate file labeled 'Manuscript'.

We look forward to receiving your revised manuscript.

Kind regards,

Emiliano Cè, Ph.D.

Academic Editor

PLOS ONE

**Journal Requirements:**

Reviewers' comments:

Reviewer's Responses to Questions

**Comments to the Author**

Reviewer #1: All comments have been addressed

Reviewer #2: All comments have been addressed

2. Is the manuscript technically sound, and do the data support the conclusions?

Reviewer #1: Yes

Reviewer #2: Yes

3. Has the statistical analysis been performed appropriately and rigorously?

Reviewer #1: Yes

Reviewer #2: Yes

4. Have the authors made all data underlying the findings in their manuscript fully available?

Reviewer #1: Yes

Reviewer #2: No

5. Is the manuscript presented in an intelligible fashion and written in standard English?

Reviewer #1: Yes

Reviewer #2: Yes

**Reviewer #1:**  (No Response)

**Reviewer #2: ** Regarding the image resolution, I would like to emphasize that this issue should be decided entirely by the editor and the editorial office.

No further reviewer assessment is necessary on this point.

Figure resolution is one of the most fundamental aspects of publication quality, and in my opinion, the current quality is not suitable for publication.

I cannot take responsibility if the paper is published as it is and later criticized for poor figure quality.

Apart from this issue, I have no objection to the rest of the manuscript.

**Do you want your identity to be public for this peer review?** For information about this choice, including consent withdrawal, please see our Privacy Policy

Reviewer #1: No

Reviewer #2: No

---

## [Author Response · Author response to Decision Letter 3]

3 Nov 2025

Dear Dr. Emiliano Cè,

Following the second revision, we would like to sincerely thank you once again for your valuable comments. We have made the necessary revisions to our manuscript (PONE-D-25-30664R2), entitled “Comparison of lower limb muscle activation between slow and fast tempos during the stepping test in young men.”

Regarding the comment from Reviewer #2, the only remaining issue concerned the image resolution. To address this, we used PLOS’s free online tool NAAS to verify and adjust the figures to meet the publication-quality standards. Both Figure 1 and Figure 2 were confirmed by NAAS to fully comply with PLOS ONE’s figure specifications. In addition, we confirmed that when viewing the PDF version of the manuscript, clicking the “Click here to access/download” link in the upper right corner of each figure opens the original image, which appears clear and high in resolution.

As Reviewer #2 kindly mentioned, the evaluation of image resolution should be handled by the editor and the editorial office. Therefore, we would greatly appreciate it if you could review the figures and confirm that they meet the journal’s requirements. If there are still any issues with the resolution of Figures 1 or 2, please kindly let us know.

As mentioned above, since the revision concerned only the figure resolution and no changes were made to the manuscript text itself, the “Manuscript” and “Revised Manuscript with Track Changes” files are identical to those submitted in the second revision. Therefore, there are no highlighted modifications in the tracked-changes version.

Thank you very much for your continued consideration.

Warm regards,

Keiji Koyama

---

## [Decision Letter · Decision Letter 3]

6 Nov 2025

Comparison of lower limb muscle activation between slow and fast tempos during the stepping test in young men

PONE-D-25-30664R3

Dear Dr. Koyama,

We’re pleased to inform you that your manuscript has been judged scientifically suitable for publication and will be formally accepted for publication once it meets all outstanding technical requirements.

Kind regards,

Emiliano Cè, Ph.D.

Academic Editor

PLOS ONE

Additional Editor Comments (optional):

Dear Authors,

both the reviewers have required to strongly improve the quality of the Figures as is seems to be too low in the present version.

Please, consider this point during the revision of the proof.

Best regards

Reviewers' comments:

Reviewer's Responses to Questions

**Comments to the Author**

Reviewer #1: (No Response)

Reviewer #2: All comments have been addressed

2. Is the manuscript technically sound, and do the data support the conclusions?

Reviewer #1: Yes

Reviewer #2: Yes

3. Has the statistical analysis been performed appropriately and rigorously?

Reviewer #1: Yes

Reviewer #2: Yes

4. Have the authors made all data underlying the findings in their manuscript fully available?

Reviewer #1: Yes

Reviewer #2: No

5. Is the manuscript presented in an intelligible fashion and written in standard English?

Reviewer #1: Yes

Reviewer #2: Yes

Reviewer #1: The authors are requested to replace Figures 1 and 2 with higher-resolution versions (at least 300 dpi), as this is a REQIRNMENT for final acceptance. Despite previous requests from the reviewer, the figures have not yet been updated to meet this standard.

Reviewer #2: (No Response)

**Do you want your identity to be public for this peer review?** For information about this choice, including consent withdrawal, please see our Privacy Policy

Reviewer #1: No

Reviewer #2: No

---

## [Editor Report · Acceptance letter]

PONE-D-25-30664R3

PLOS ONE

Dear Dr. Koyama,

I'm pleased to inform you that your manuscript has been deemed suitable for publication in PLOS ONE. Congratulations! Your manuscript is now being handed over to our production team.

Kind regards,

on behalf of

Prof. Emiliano Cè

Academic Editor

PLOS ONE